# Antibiotic-Free Poultry Meat Consumption and Its Determinants

**DOI:** 10.3390/foods12091776

**Published:** 2023-04-25

**Authors:** Hosein Mohammadi, Sayed Saghaian, Flavio Boccia

**Affiliations:** 1Department of Agricultural Economics, College of Agriculture, Ferdowsi University of Mashhad, Mashhad 9177948978, Iran; 2Department of Agricultural Economic, College of Agriculture, Food and Environment, University of Kentucky, Lexington, KY 40536, USA; ssaghaian@uky.edu; 3Department of Economic and Legal Studies, Parthenope University of Naples, 80132 Naples, Italy; flavio.boccia@uniparthenope.it

**Keywords:** antibiotic-free poultry, meat, food security, marketing, sustainable consumption

## Abstract

In recent decades, meat consumption has increased globally due to increased incomes. A consequence of increased red meat consumption has been the rise in greenhouse gas emissions and nutrition-related diseases. Consumption of antibiotic-free (ABF) poultry meat is a viable healthy and sustainable substitute that will cause less damage to humans and the environment in the long run. This study was undertaken due to the increasing importance of healthy food consumption to preventing nutrition-related diseases. The health food industry is still in its preliminary stages; for market development of organic broiler meat and movement toward sustainable production of ABF meat, the first necessary step is conducting empirical research on ABF poultry meat consumption and identifying factors that influence household consumption patterns of ABF poultry meat. Therefore, the objectives of this study were the investigation of factors affecting poultry meat consumption by consumers and ABF poultry meat preference. Comparing the results could reveal what percentage of consumers are able to buy healthier higher-priced antibiotic-free poultry meat. Data were collected from 360 completed questionnaires completed by households from the city of Mashhad, Iran via simple random sampling in 2021. To investigate the first objective, an ordered logit model was used. The results showed that age, the head of household’s education, awareness of the nutritional benefits of poultry meat, advertising, and family income were statistically significant determinants of poultry meat consumption. To investigate the second objective, since some consumers cannot buy ABF poultry meat due to the higher prices, we used a two-step Heckman model. The results showed that the awareness of the nutritional benefits of ABF poultry meat, the head of household’s education, monthly family income, and advertising had positive impacts, with prices having a negative impact on the amount of antibiotic-free poultry meat consumed by the households. Comparing the results of the two models revealed that only about 30% of consumers could buy ABF poultry meat, mainly due to the higher prices. This study recommends improving consumer awareness, targeted distribution of ABF poultry meat according to customers’ economic and demographic characteristics, affordable prices, and appropriate marketing tools for sustainable consumption of ABF poultry meat.

## 1. Introduction

Nutritionists and informed consumers are concerned about food security, with this concern including the potential risks from consuming animal-based food products [1]. Food security has received much attention and has been a key concern of the Food and Agriculture Organization (FAO) of the United Nations. The FAO [2] definition of food security involves physical, social, and economic access to an adequate and safe supply of nutritious food to maintain an active and healthy life. Due to increasing per capita income and population growth, the average consumption of meats, which are an important source of nutrients, has increased globally [3]. However, the consumption of different types of meats has significant impacts on people’s health and the environment [4]. The literature shows that production of 1 kg of protein from beef requires 10 times more water, 12 times more fertilizer, 18 times more land, 10 times more pesticides, and 9 times more fuel compared to the same amount of protein obtained from kidney beans [5]. Although beef has 25.4 g of protein per 100 g and kidney bean has 8.1 g of protein per 100 g. Livestock production also impacts greenhouse gas emissions, the water-use footprint, water pollution, and water scarcity negatively [6]. Hence, there is a need to change current lifestyles and consumption habits. 

Many consumers are seeking alternatives to conventional meat products that are produced without routine use of antibiotics; this change is driven by consumer demand for meats without antibiotics. In general, poultry meat contains lower amounts of environmental pollutants [6]. Rising consumer interest in safer food production has led to the growth of antibiotic-free (ABF) meat production; several producers and retail-chains have moved their marketing in that direction [7]. Consumer demand for ABF broiler meat is increasing; however, there are challenges in adopting suitable strategies to produce ABF broiler meat [8]. 

According to the United States Organic Trade Association, sales of antibiotic-free and organic foods have increased by 20% per year since 1990; however, there are numerous challenges faced by companies attempting to produce ABF poultry meat, such as production, management, health and animal welfare challenges [9].

ABF poultry meat production is becoming increasingly popular due to consumers’ emerging perceptions of it as superior to conventional broiler meat. In the poultry sector, ABF farms are increasing in number and addressing consumer concerns and new EU regulations [10]. This trend is motivated by concerns over personal health, the environment, animal welfare, taste, and quality. The elimination of antibiotics in raising broilers can have a positive effect on the conservation of natural resources; however, it may also have negative economic effects via increased costs of production and higher prices for consumers. 

Animal feeding regimes and nutrition are important in the livestock industry; however, higher than standard doses of antibiotics and artificial growth stimulants in feed can contribute to a build-up of residue in meat, animal products, and the environment (water and soil pollution) [11].

The Centers for Disease Control and Prevention (CDC) and the FAO have declared antimicrobial resistance as a serious threat to world health [12,13]. Antibiotic residue in meat and other foods facilitates allergic reactions in many people and can weaken the human immune system [14]. Therefore, reducing unnecessary use of antibiotics is important; an important target in this field is to reduce applications of antibiotics in animals for human consumption, such as cattle, pigs, and poultry [15]. 

There is a growing movement towards production and consumption of ABF products and consumer demand is rapidly growing for meat raised without routine use of antibiotics [16]. According to the definition devised by the U.S. Department of Agriculture, Food Safety, and Inspection Service, (USDA-FSIS), the label “antibiotic-free poultry” means that no antibiotic drug has been used in the food or water consumed by the fowl and no antibiotics have been injected into it [9]. Antibiotics are used to prevent, treat, and control bacterial infections in poultry and livestock. In addition, antibiotics can increase animal performance; thus, by using them, farmers can produce more meat with less feed [17]. Over the years, antibiotics have been used in the poultry industry to prevent and treat diseases and promote growth. ABF poultry production is now a common trend worldwide and has many categories according to type of certification, market claim, and consumer product [18]. 

The main ABF groups include standard ABF programs that allow use of chemical antibacterial, chemical anticoccidials, and ionophores, but no antibiotics. Organic production allows no antibiotics, ionophores, coccidiostats, chemical anticoccidials, or chemical antibacterial products. Therefore, antibiotic-free poultry is somewhat different from organic poultry because only the use of antibiotics is prohibited in ABF poultry, whereas antibiotics are prohibited in organic poultry and no other chemical additives are allowed in their water or poultry feed throughout their growth period. Thus, organic is ABF, but not all ABF is organic. Moreover, organic broiler meat has variable costs that are 70% higher and fixed costs that are 86% higher than conventional production. This higher production cost for organic occurs due to feed, labor, certification, and outdoor maintenance costs [19]. However, the price of organic broiler meat is double the price of conventional broiler meat and, thus, organic broiler meat production is more profitable than conventionally reared production. 

Nowadays, it seems that the consumption of organic and healthy products is increasing in major cities, albeit mostly by high-income people [20]. There is a significant tendency to purchase fresh poultry of domestic and organic origin because of its quality and safety [21]. These results are particularly useful for product marketing and future product development in the locally sourced and produced poultry sector. Consumers are looking for high-quality, healthy products; however, consumption by the public is low mostly because of high prices for such products, though other reasons related to flawed distribution and marketing systems also play a role [22]. In addition to the difficulty of marketing these products, the main barrier to more widespread consumption is a lack of knowledge and limited information on their benefits [23]. Generally, ABF farming should be standardized for protect both animal welfare and consumers’ health [24]. 

The aim of this research was to investigate the factors affecting poultry meat consumption and then investigate the factors affecting consumption of ABF poultry meat. Since the factors affecting the consumption of poultry meat may be different from the factors affecting the consumption of ABF poultry meat, in this study two different models were used to investigate these two issues. Previous studies have investigated factors affecting consumption of organic or healthy products in the agricultural sector. For example, previous studies identified socio-economic factors, such as education and age [25], gender and occupation [26], price and income [27], product quality [28], and consumer awareness and environmental considerations [29]. However, there is no study with the methodology employed in this research related to the consumption of ABF poultry meat. 

According to FAO (2022), global poultry meat production in 2022 is more than 138.8 million tones; however, a decline in poultry meat production in Iran is anticipated due to diminishing profit margins for poultry meat producers amid high feed prices and farm-gate price caps imposed by the government to control food price inflation [30]. The production of poultry meat in Iran in 2022 is estimated to be more than 2.1 million tons; Iran likely consumed about 2.2 million tons of poultry in the same year [30]. Therefore, the per capita consumption of poultry meat in Iran in 2022 is estimated to be around 26 Kg. Mashhad is the capital of the Khorasan–Razavi province, which is the third most important province in Iran for poultry meat production, producing around 120 thousand tons of poultry meat per year. Per capita consumption of poultry meat in the province is about 30 Kg per year [31], considering the import and export of poultry meat to the province from other provinces. Moreover, around 10 % of the poultry produced in this province is antibiotic-free; this percentage may increase with consumer demand for ABF poultry increasing. 

In the next section, we present the details of the methodology and methods of estimation; in later sections, the results, discussions, and policy implications are presented. 

## 2. Materials and Methods

This study aimed to first identify and evaluate factors affecting the consumption level of poultry meat among households in Mashhad, Iran, using ordered logit model. In the second stage, we investigated the factors affecting the consumption of ABF poultry meat using a two-step Heckman Tobit regression model.

### 2.1. Ordered Logit Model

An ordered logit model (OLM) was employed to analyze the factors affecting the level of poultry meat consumption among the households. OLM is based on a latent variable and, in this study, it constituted the amount of poultry meat consumed by households. This latent variable is defined as follows:(1)$$y_{i}^{*}=X_{i}\beta +\varepsilon_{i}\;-\infty <y_{i}^{*}<\infty$$
where $y_{i}^{*}$ is continuous latent variable that underlies the observed ordinal data; here, it is the per capita consumption of poultry meat by a household per month. β is a vector of estimated parameters, $X_{i}$ is a vector of explanatory variables, and $\varepsilon_{i}$ is an error term that has a logistic distribution. Since the amount of $y_{i}^{*}$ is not observable, the standard regression techniques, such as Ordinary Least Square (OLS), are not applicable [32]. If $y_{i}$ is a discrete and observable variable that measures the level of consumption, the relationship between unobservable variable $y_{i}^{*}$ and $y_{i}$ can be obtained via the ordered logit model as follows:(2)$$\begin{array}{l} y_{i}=1\;if\;-\infty \leq y_{i}^{*}<\tau_{1}\;i=1,\ldots ,n\\ y_{i}=2\;if\;\tau_{1}\leq y_{i}^{*}<\tau_{2}\;i=1,\ldots ,n\\ y_{i}=3\;if\;\tau_{2}\leq y_{i}^{*}<\tau_{3}\;i=1,\ldots ,n\\ y_{i}=4\;if\;\tau_{3}\leq y_{i}^{*}<\tau_{4}\;i=1,\ldots ,n\\ ,\ldots \end{array}$$

In these relationships, *n* is sample size and τ’s are thresholds that define observable discrete responses that should be estimated. The probability that $y_{i}=J$ can be calculated as depicted in Equation (3):(3)$$\mathrm{Pr}(y_{i}=j)=\mathrm{Pr}(\varepsilon_{i}<\tau_{j}-X_{i}\beta |X_{i})-\mathrm{Pr}(\varepsilon_{i}<\tau_{j-1}-X_{i}\beta |X_{i})=F(\tau_{j}-X_{i}\beta |X_{i})-F(\tau_{j-1}-X_{i}\beta |X_{i})$$

In Equation (3), F is a cumulative distribution function (CDF) for ε, β is a column vector of parameters, and $X_{i}$ is a vector of independent variables. One important test in OLM is the assumption of parallel regression; thus, we tested whether the βs were equal for each level of dependent variable. 

Model (3) was estimated via using maximum likelihood (ML) method; marginal effects of each variable were calculated at the means of independent variables, with the summation of marginal effects for each variable being zero. The marginal change in the probability can be computed as follows:(4)$$\frac{\partial \mathrm{Pr}(y=m|X)}{\partial x_{k}}=\frac{\partial F(\tau_{m}-X\beta )}{\partial x_{k}}-\frac{\partial F(\tau_{m-1}-X\beta )}{\partial x_{k}}$$

The real amount of poultry meat consumed by a household is a continuous variable. Most consumers did not know the exact amount of their per capita monthly consumption; thus, poultry meat consumption was categorized according to three levels. The three categories for consumption of poultry were established according to monthly minimum and maximum per capita consumption in the sample and the standard deviation of data. Households with per capita consumption between 0.5 to 1.33 Kg per month were classified as low-level consumers, while households with per capita consumption between 1.33 to 3.16 Kg per month were classified as medium-level consumers; households with per capita consumption of 3.16 to 5 Kg per month were considered high-level consumers. In addition, after specifying the model, variables of age and education level of the household head, monthly family income, price paid for poultry meat, price paid for red meat, advertising, awareness of the nutritional benefits of poultry, and the number of children under 10 years old in the family were selected as variables to explain the ordinal regression.

### 2.2. The Tobit Model

The Tobit model was used with the Heckman’s two-step model to investigate factors influencing consumption of antibiotic-free poultry meat. The Tobit model was applied because logit and probit models cannot distinguish between factors that influence decisions to consume ABF poultry meat, as well as factors that influence level of consumption of ABF poultry meat. The structure of the Tobit model is expressed as follows:(5)$$\begin{array}{l} Y_{i}=\beta^{\prime }X_{i}+U_{i}\;Y_{i}^{*}>0\\ Y_{i}=0\;Y_{i}^{*}\leq 0\\ i=1,\ldots ,n \end{array}$$
where $Y_{i}^{*}$ is the latent variable, $Y_{i}$ is the observed variable, $\beta^{\prime }$ is the vector of model parameters, $X_{i}$ is the vector of independent variables, $U_{i}$ is the disturbance term, and n is the total number of observations [33]. For observations of antibiotic-free poultry consumption, $Y_{i}^{*}$ is consumption level, while for observations that antibiotic-free poultry are not consumed, $Y_{i}^{*}$ is zero. Thus, the cutting threshold was zero.

The Tobit model utilized observations of potential and actual consumers to resolve Type I error (non-random sampling). Nevertheless, it did include the risk of Type II error, the lack of differentiation between the factors affecting decisions to consume, and factors affecting consumption level. Heckman suggested a two-step method for solving the second problem. Heckman’s two-step method assumes that one set of variables affects the decision to engage in a specific activity, while another set of variables affects the volume of participation in the activity [34].

Accordingly, the first step is estimating a model that shows the probability of consuming antibiotic-free poultry meat and, for this part, the probit regression model was used as below: (6)$$\begin{array}{l} Z_{i}=\beta^{\prime }X_{i}+v_{i}\;i=1,2,\ldots n\\ Z_{i}=1\;if\;Y_{i}^{*}>0\\ Z_{i}=0\;if\;Y_{i}^{*}<0 \end{array}$$

$Z_{i}$ is the dependent variable of the first step. If a household consumes antibiotic-free poultry meat, its value is 1; otherwise, its value is zero. The first step estimates factors affecting a household’s decision to consume antibiotic-free poultry meat. The inverse Mills ratio (IMR), $\lambda =\frac{\phi (\beta^{\prime }X_{i})}{\varphi (\beta^{\prime }X_{i})}$ is the ratio of the standard normal density function to the standard normal cumulative distribution function. 

In the second step, the relationship between the independent variables and antibiotic-free poultry meat consumption is estimated using observations of $Y_{i}$ on $X_{i}$ and IMR, which are obtained from the first step of probit analysis: (7)$$Y_{i}=\beta^{\prime }X_{i}+\sigma IMR_{i}+e_{i}$$

The second estimation shows how the explanatory variables affect consumption levels for antibiotic-free poultry meat. The IMR coefficient measures errors resulting from sampling and, if they is significantly different than zero, it indicates bias in the sampling [35]. The presence of the inverse Mills ratio variable in the above linear regression model removes the variance heteroscedasticity of the initial model and permits use of the ordinary least squares estimator [36]. 

Data for this study were collected from questionnaires completed by 360 household heads in the city of Mashhad, Iran, during 2021. The population of Mashhad was approximately 3.2 million people in 2021; the sample size was determined in accordance with Cochran’s formula for a representative sample [37]. The respondents were the main person making household decisions on purchases of poultry meat. The questions that were asked of the household heads by face-to-face interviews are classified into two general groups. A series of questions covered the general information of the household, such as the number of children, gender and age of the head of household, the presence of children or elderly people in the household, the monthly amount of poultry meat consumption, and the monthly income of the household. The second group of questions covered the effect of the variables of the price of poultry and alternative meats on the level of poultry meat consumption by households and the effect of quality, advertising, and awareness of the benefits of poultry meat on the level of household poultry meat consumption; this group also covered the effect of awareness of the benefits of antibiotic-free poultry meat on the level of household consumption. 

## 3. Results

Information about research variables and descriptive statistics of data are presented in Table 1. The explanatory variables collected from the survey include gender, age and level of education of the household head, family income, number of elderly (over 60 years) people in the household, quality of the product, awareness of nutritional benefits, prices, and advertisements. Some studies have shown that demographic variables, such as age, gender, presence of young children in the family, family size, education, income, and experience, impact decisions to purchase green food products [38]. 

The survey results indicated that 55% of the households were low-level poultry consumers, while 18% were medium-level and 28% were high-level consumers. Only 33% of the sample purchased ABF poultry in 2021. 

According to the information obtained through the questionnaire, the reasons for non-consumption of antibiotic-free poultry meat were higher prices (18%), lack of information and knowledge about the product (43%), shortage of adequate centers that supply the product (36%), and the low weight of antibiotic-free poultry meat in comparison to traditional poultry meat (3%). Households that consumed antibiotic-free poultry meat paid more attention to price (25%), packaging (10%), brand (38%), weight (11%), and source of supply (16%). Table 1 shows descriptive statistics of the explanatory variables in both Ordered and Tobit regression models.

### 3.1. Ordered Logit Model Results

Table 2 presents results of the parallel regression test in the ordered logit model. The null hypothesis is not rejected, meaning that the ordered logit model is a suitable model that shows the relationships between explanatory variables and groups of dependent variables.

Table 3 presents results of the ordered logit model. In this model, the dependent variable is based on household monthly per capita consumption of poultry meat according to three categories: low, medium, and high consumption. According to Pseudo R-squared statistics, the estimated logit model has an appropriate level of goodness of fit and the independent variables explain a suitable degree of the consumption level of poultry meat. The likelihood ratio (LR) chi-square test results reject the hypothesis that all coefficients except the intercept are zero.

Results presented in Table 3 indicate that the age and education level of the household head, awareness of the nutritional benefits of poultry meat, and advertising all have a positive effect on the probability that a household had a higher per capita consumption level of poultry meat. The monthly household income and the poultry price have an adverse effect on the probability of poultry consumption, though only family income was statistically significant. Increasing family income probably allows households to substitute higher-priced meat for poultry. In [39], the authors showed that the relatively low and competitive price of poultry compared to other meat, nutritional qualities, and the absence of cultural or religious obstacles, are the main factors explaining poultry meat attractiveness.

The coefficients in Table 3 only show direction effects. Marginal effects with respect to each explanatory variable for each level of the dependent variable are shown in Table 4. Variables such as awareness of the benefits of certain meat products and advertisements are dichotomous variables; when these variables changed (with all other variables held at their mean constant), the possibility that the dependent variable will stand at each of the three levels also changed, as indicated in Table 4.

The results in Table 4 indicate that if the household sees advertisements, the probability that it will be in group 1 (low consumption) reduces by 0.08%, the probability that it will be in group 2 (medium consumption) increases by 0.03%, and the probability that it will be in group 3 (high consumption) increases by 0.05%. Advertisement in any form increases the consumption level of poultry meat. For continuous variables, such as education, if the education level of household head increased over one year, the probability that it will be in group 1 (low consumption) reduced by 0.16%, while the probability that it will be in group 2 (medium consumption) or group 3 (high consumption) increased by 0.06 and 0.1% respectively. With an increase in education, people were more willing to consider nutrition [39]. They paid more attention to the properties of white meat compared to red meat; thus, consumption increased. Other variables were interpreted in the same way. 

### 3.2. Probit Model Results

The results of the probit model for determining factors that effectively influence consumption of antibiotic-free poultry meat are reported in Table 5. The results show that level of education household head, family income, quality of antibiotic-free poultry meat, awareness of the benefits of antibiotic-free meat, and advertisement had significant effects on consumption of antibiotic-free poultry meat. Education, income, quality, awareness of benefits, and advertisement had direct and positive effects on consumption; however, age and price had indirect and negative yet insignificant effects on consumption of antibiotic-free poultry meat. Similar studies also showed positive relationships between income and educational levels with consumption of green food products [40]. Furthermore, commercial strategies, such as advertising poultry products, affected the extent of their consumption [39]. 

According to results of the estimated probit model in Table 5, by holding other variables constant, with an increase in the level of education and family income, the probability of a decision to consume antibiotic-free poultry meat increased by 0.014 and 0.007%, respectively. Highly educated men were the main consumers of pollution-free foods, proving that improved living standards and a rising middle class can increase demand for green food products [38,41]. In fact, it can be said that people with a higher level of education demand more healthy products as part of their daily diet. In addition, a higher household income will increase purchasing power for antibiotic-free poultry meat; thus, consumption can increase accordingly. Moreover, low demand for organic products and sales problems were the most significant impediments to producing antibiotic-free chicken; there was a positive relationship between attitude to production of antibiotic-free chicken and level of education [42].

In addition, results presented in Table 5 show that more advertising will increase the probability of a decision to consume antibiotic-free poultry meat by 0.04%. The authors of [38] also showed that advertising enhanced consumption of poultry meat. Furthermore, the quality of a product and awareness of the nutritional benefits of antibiotic-free poultry meat had a significant effect on decisions to consume antibiotic-free poultry meat. In [37], it was shown that problems with consumption of antibiotic-free food products, such as confidence in product safety, storage conditions, quality of product, the production process, and accessibility as well as credibility of product information, influence purchasing decisions. Moreover, they suggested effective strategies, such as distributing information about green products to consumers, to develop trust in the product. The other explanatory variables had no significant effect on the dependent variable. The value of Pseudo R-squared was 0.25, which indicates goodness of fit of the probit model and represents appropriate fitness of the model. 

Table 6 presents results of estimates of the linear regression model. As can be seen, increased levels of education for the head of household and family income increased purchases of antibiotic-free poultry meat by 0.23 and 0.07%, respectively. Furthermore, the estimated coefficients for advertisements and awareness of nutritional benefits of antibiotic-free poultry meat had a positive and significant effect; an increase in the level of each of these variables increased purchase rates of antibiotic-free poultry meat by 0.27 and 0.9%, respectively. 

Moreover, the price of the product had a negative and significant effect on consumption of antibiotic-free poultry meat, such that if the price increased by one unit, then purchases decreased by 0.19%. In [38], it was shown that a higher price was a major factor affecting growth in consumption of green products because there was a large price differential between conventional and green food products. 

The last variable in Table 6 is the inverse Mills ratio, which was not significant and demonstrates no difference between the variables effective on decisions to consume antibiotic-free poultry and the rate of antibiotic-free poultry consumption. Therefore, removing zero observations did not cause bias and only reduced efficiency of the estimators.

## 4. Discussion

Many consumers across the world are concerned about food safety. This concern includes the potential risks of consuming animal-based food products. Food safety and security has received much attention and has been one of the main concerns of the Food and Agriculture Organization (FAO). Food security includes physical, social, and economic access to an adequate and safe supply of nutritious food to maintain an active and healthy lifestyle. The most common health problems worldwide are caused by diseases related to food contamination, which impose limitations on economic productivity, especially in developing countries. Due to the increase in per capita income and population growth, the average consumption of meat has increased globally. The consumption of various types of meat has a significant impact on the health of people and the environment.

Many consumers are looking for alternatives to conventional meat products that are produced without the routine use of antibiotics; these changes are drivers of consumer demand for antibiotic-free meats. There is a growing movement towards the production and consumption of ABF products, and consumer demand for meat produced without the routine use of antibiotics is growing rapidly. Poultry meat production is more sustainable in the long run compared to red meat production in meeting consumer protein needs. In addition, the production of ABF poultry meat is a response to environmental considerations. Increased consumer interest in safer food has led to a growth in ABF meat production; several producers have shifted their marketing towards this direction. While consumer demand for ABF broiler meat is increasing, there are challenges in adopting appropriate strategies for ABF meat production. 

This study contributes to the literature by examining the effects of marketing factors, such as advertising, along with the effects of consumer-related factors, such as consumer awareness, in the framework of a Tobit model to investigate factors affecting the consumption of ABF poultry meat. The Tobit model is applied because other related models cannot distinguish between the factors that influence decisions to consume ABF poultry meat and the factors that influence level of consumption of ABF poultry meat. In this study, we have addressed this research gap in ABF poultry meat. For this purpose, two groups of factors—marketing and consumer characteristics—were used and their effect on poultry meat consumption and consumption of ABF poultry meat was investigated.

The survey results of the questionnaires showed that 33% of households consumed antibiotic-free poultry meat. The lack of information about the product and shortage of adequate centers supplying the product were the main reasons for the low consumption levels. Moreover, brand, price, and supply centers were important factors for those households that consumed antibiotic-free poultry meat. 

The results of the ordered logit model indicated that the age and education level of the household head, awareness of the nutritional benefits of poultry meat, and advertising all had a positive effect on the probability that a household had a higher per capita consumption level of poultry meat. Monthly household income and the price of poultry had an adverse effect on probability of poultry consumption, though only family income was statistically significant. Increasing family income probably allows households to substitute for higher-priced ABF poultry meat.

The results of the probit model, which was to determine factors influencing consumption of antibiotic-free poultry meat, showed that the level of education household head, family income, quality of antibiotic-free poultry meat, awareness of the benefits of antibiotic-free meat, and advertisement had significant effects on consumption of ABF poultry meat. Education, income, quality, awareness of benefits, and advertisement had direct and positive effects on consumption; however, age and price had indirect negative but insignificant effects on consumption of ABF poultry meat. Highly educated men were the main consumers of pollution-free foods. Improved living standards and a rising middle class increase demand for green food. Moreover, low demand for organic products and sales problems were the most significant impediments to producing ABF chicken. There was a positive relationship between the attitude to production of ABF chicken and level of education. 

Furthermore, the estimated coefficients of the linear regression model showed that advertisement and awareness of nutritional benefits of ABF poultry meat had a positive and significant effect; however, the price of the product had a significant negative effect on consumption of ABF poultry meat; there was also a large price differential between conventional and green food products. Hence, targeted market segmentation and wider distribution of antibiotic-free poultry meat among the wealthy and more educated consumers in the society can serve boost production and enhance consumption of ABF products. Due to the limitations of this study, the random sampling method was used to collect data. Considering the different economic and social characteristics of different urban areas and different cities, it is better to use other sampling methods, such as random classification sampling, in order to more closely examine the effects of differences in urban areas on the consumption pattern of poultry meat.

## 5. Conclusions

According to the results of this study, it can be concluded that a significant percentage of households do not consume antibiotic-free poultry meat because of unfamiliarity with the product and lack of information on its nutritional benefits. With increased consumption and demand for the product, its production will also increase. Improving the awareness of antibiotic-free poultry meat and advertisement had a positive and significant effect on the level of consumption. Hence, policymakers focusing on increasing the production of antibiotic-free and healthy products need to consider the factors that affect consumption of these products, including the use of appropriate marketing tools. It is recommended that by using appropriate promotional tools, government and NGOs that promote production and consumption of antibiotic-free products create programs that increase consumer awareness of the health and environmental benefits of antibiotic-free products, as well as the detrimental effects of products containing antibiotics. 

Furthermore, the supply center shortages of antibiotic-free products had a significant effect on the low-level consumption of antibiotic-free poultry meat. Therefore, targeted distribution of and sales centers for antibiotic-free products could increase sales and consumption. In addition, prices played an important role in determining the consumption level of antibiotic-free poultry meat. Efforts to supply antibiotic-free products with reduced production costs could lead to lower prices, while other policies, such as brand development, could have a positive effect on the consumption of antibiotic-free products. Having more brands entering the market and brand competition could improve not only the quality of antibiotic-free meat products, but also decrease prices, which may in turn contribute to an increase in consumption levels. 

## Figures and Tables

**Table 1 foods-12-01776-t001:** Descriptive statistics for explanatory variables.

Variable	Mean	SD	Min	Max	Description
Gender of household head	0.92	0.12	0	1	Women = 0 and men = 1
Age of household head	43.9	6.5	21	80	Years
Education of household head	14	2.5	0	22	Years
Monthly income	120	23	18	350	Iranian million Rials
Price of poultry meat	0.62	0.32	0	1	Low = 0, high = 1
Price of alternative meat	0.67	0.38	0	1	Low = 0 and High = 1
Quality	0.56	0.48	0	1	Low = 0 and High = 1
Advertising	0.63	0.5	0	1	Low = 0 and High = 1
Awareness of poultry meat benefits	0.67	0.29	0	1	Low = 0 and High = 1
Awareness of ABF meat benefits	0.55	0.31	0	1	Low = 0 and High = 1
Children below 10	2.2	1.5	0	5	Count
Elderly member(s) over 60	1.3	0.7	0	3	Count

**Table 2 foods-12-01776-t002:** Results of the parallel regression test.

Test	Chi-Square	df	Prob.
Brant	10.8	8	0.213

**Table 3 foods-12-01776-t003:** Ordered logit regression model results.

Variable	Coefficient	SD	Z Statistic	Prob.
Age	0.37 *	0.019	1.92	0.055
Education	0.72 **	0.34	2.1	0.036
Income	−0.63 *	0.34	−1.82	0.07
Poultry meat price	−0.11	0.24	−0.46	0.65
Alternative meat price	−0.09	0.21	−0.43	0.67
Awareness of meat benefit	0.38 *	0.23	1.67	0.09
Advertisement	0.39 **	0.19	2.05	0.04
Children	0.1	0.49	0.2	0.83
*τ* _1_	4.85 **	1.89		
*τ* _2_	5.79 **	1.91		
Log likelihood = −84.7	LR chi^2^(10) = 15.7			
$Pseudo\;R^{2}=0.18$	Prob. > chi^2^ = 0.04			

*, ** indicates 10% and 5% significance level respectively.

**Table 4 foods-12-01776-t004:** Marginal effects of ordered logit model.

Marginal Effect of Dependent Variable	Group 1 (Low Consumption)	Group 2 (Medium Consumption)	Group 3 (High Consumption)
Age	−0.0085	0.0031	0.0054
Education	−0.16	0.060	0.1
Income	0.14	−0.05	−0.09
Poultry meat price	0.026	−0.01	−0.016
Alternative meat price	0.022	−0.008	−0.014
Awareness of meat benefit	−0.09	0.03	0.06
Advertisement	−0.08	0.03	0.05
Children	−0.02	0.008	0.014

**Table 5 foods-12-01776-t005:** The results of probit model for determining factors affecting ABF poultry consumption.

Variable	Coefficient	Z Statistic	Prob.	Marginal Effect
Gender	0.24	0.39	0.695	0.072
Age	−0.024	−1.50	0.13	−0.008
Education	0.43 *	1.79	0.08	0.014
Income	0.21 *	1.91	0.06	0.007
Price	−0.09	1.41	0.15	−0.03
Quality	0.37 **	2.06	0.04	0.12
Awareness of ABF meat benefits	0.47 ***	2.98	0.003	0.15
Advertisement	0.12 *	1.66	0.09	0.04
Elderly member	−0.15	1.42	0.14	−0.41
Log likelihood = −47.83		LR chi^2^(10) = 24.28		
$R_{McF}^{2}=0.25$		Prob. > chi^2^ = 0.03		

*, **, *** indicates 10%, 5% and 1% significance level respectively.

**Table 6 foods-12-01776-t006:** The results of second stage of Heckman model for determining factors affecting the amount of antibiotic-free poultry meat consumption.

Variable	Coefficient	t Statistic	Prob.
Gender	0.082	0.22	0.824
Age	−0.035	−1.44	0.16
Education	0.23 *	1.67	0.09
Income	0.07 *	1.79	0.07
Poultry meat price	−0.19 *	−1.81	0.07
Quality	0.66	1.04	0.24
Awareness of ABF meat benefit	0.9 **	2.3	0.03
Advertisement	0.27 **	2.2	0.03
Elderly member	0.73	1.50	0.14
Inverse Mills ratio	1.7	1.57	0.12

*, ** indicates 10% and 5% significance level respectively.

## Data Availability

Not applicable.

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
