# Peer review of "Antibiotic-Free Poultry Meat Consumption and Its Determinants"

_foods, 2023, doi:10.3390/foods12091776_

Round 1

Reviewer 1 Report (New Reviewer)

Dear authors:

I consider that this is a valuable document because it will help antibiotic-free poultry meat producers to determine demographics and range of consumers they might be targeting. The article is well structured, the writing is good, and it is quite understandable. In order to improve it, I have write some suggestions below, that I hope that could be useful for the authors (I have annexed a Word document).

Antibiotic-free poultry meat consumption and its determinants

Dear authors:

I consider that this is a valuable document because it will help antibiotic-free poultry meat producers to determine demographics and range of consumers they might be targeting. The article is well structured, the writing is good, and it is quite understandable. In order to improve it, I have write some suggestions below, that I hope that could be useful for the authors.

Lines 45-48: Could you please mention a brief comparison of the protein content that 100 grams of beans contain compared to 100 grams of beef? I feel that when comparing them, the statement of these lines may not be true, because as I understand it, 100 g of beef contain approximately 16-22% protein, while beans have between 8-11%. Please check it.

Line 74: Please add the missing full stop at the end of the sentence.

Line 76: Please erase the space between references 12, and 13.

Line 76: Please add, instead of “the authors of [14]”, the last name of the authors.

Line 86: I think that it would be better to use “fowl” or “poult” instead of “bird”.

Line 87: Please erase the extra space after “[17].”

Line 89: Please replace the word “antibiotics” with “them”, so it doesn't sound repetitive.

Line 101: Please replace “the authors of [21]” with the author´s last names.

Lines 108, 118: Please place the last names of the authors instead of only the numbers of the citations.

Lines 135- 137 and 139-140: Please provide the reference of these data.

Lines 141-151: I believe that this paragraph should not be in this section but perhaps a part of it could be placed in the methodology and the other part in conclusions. Please reconsider it.

Lines 189, 190 and 192: Please substitute “KG” for “Kg”.

Lines 424-431: I consider that this paragraph is already mentioned on Material and Methods section, so I suggest that it could be eliminated from here and can be used to complement that section.

References: Please check that all the references are written in a correct way according to the stipulated by the journal. For example, references 3 and 5, have some writing mistakes, but please check all the list in order to correct them.

Author Response

Responses to the Comments from Reviewer #1

Comments and Suggestions for Authors

I consider that this is a valuable document because it will help antibiotic-free poultry meat producers to determine demographics and range of consumers they might be targeting. The article is well structured, the writing is good, and it is quite understandable. In order to improve it, I have write some suggestions below, that I hope that could be useful for the authors.

Comment 1: Lines 45-48: Could you please mention a brief comparison of the protein content that 100 grams of beans contain compared to 100 grams of beef? I feel that when comparing them, the statement of these lines may not be true, because as I understand it, 100 g of beef contain approximately 16-22% protein, while beans have between 8-11%. Please check it.

Response: Thanks. Using 2020 USDA data, both beef and kidney beans are high in protein. Beef has 213% more protein than kidney bean. In other words beef has 25.4g of protein per 100 grams and kidney bean has 8.1g of protein.

https://fdc.nal.usda.gov/fdc-app.html

In lines 45-48 it is stated that literature shows the production of 1 kg of protein from beef requires 10 times more water, 12 times more fertilizer, 18 times more land, 10 times more pesticides and 9 times more fuel compared to the same number of proteins obtained from kidney beans [5]. And therefore the production of beef has a significant impact on people’s health and the environment. Our main focus has not been on the protein content of these products but on the environmental consequences.

Comment 2:  Line 74: Please add the missing full stop at the end of the sentence.

Response: Done. Thanks!

Comment 3: Line 76: Please erase the space between references 12, and 13.

Response: Done. Thanks!

Comment 4: Line 76: Please add, instead of “the authors of [14]”, the last name of the authors.

Response: The sentence was corrected, but according to the format of the journal, the names of the authors cannot be written in the text of the article and the paper must be in line with the prescribed template (See the instructions for authors: https://www.mdpi.com/authors/references and https://www.mdpi.com/authors ).

Comment 5:  Line 86: I think that it would be better to use “fowl” or “poult” instead of “bird”.

Response: We have done that now in the revised manuscript. Thanks!

Comment 6: Line 87: Please erase the extra space after “[17].”

Response: Done. Thanks!

Comment 7: Line 89: Please replace the word “antibiotics” with “them”, so it doesn't sound repetitive.

Response: Done. Thanks!

Comment 8: Line 101: Please replace “the authors of [21]” with the author´s last names.

Lines 108, 118: Please place the last names of the authors instead of only the numbers of the citations.

Response: The sentence was corrected, but according to the format of the journal, the names of the authors cannot be written in the text of the article and the paper must be in line with the prescribed template (See the instructions for authors: https://www.mdpi.com/authors/references and https://www.mdpi.com/authors ).

Comment 9: Lines 135- 137 and 139-140: Please provide the reference of these data.

Response: Done. Thanks!

Comment 10: Lines 141-151: I believe that this paragraph should not be in this section but perhaps a part of it could be placed in the methodology and the other part in conclusions. Please reconsider it.

Response: Thanks, this paragraph has been removed from page 3, and moved to the conclusion section.

Comment 11: Lines 189, 190 and 192: Please substitute “KG” for “Kg”.

Response: Done. Thanks!

Comment 12: Lines 424-431: I consider that this paragraph is already mentioned on Material and Methods section, so I suggest that it could be eliminated from here and can be used to complement that section.

Response: Done. Thanks!

Comment 13: References: Please check that all the references are written in a correct way according to the stipulated by the journal. For example, references 3 and 5, have some writing mistakes, but please check all the list in order to correct them.

Response:

Thanks. All the references were checked again and the necessary corrections were made.

Again, we thank the reviewers for their thoughtful comments. The comments and suggestions have helped improve the manuscript.

Reviewer 2 Report (New Reviewer)

I think the authors need to add a literature review section to clarify the positioning of this study with respect to the extant studies. It is essential to clarify why the demographic (independent) variables were expected to influence consumer buying decisions on healthy foods.

Author Response

Responses to the Comments from Reviewer #2

Comment 1:  I think the authors need to add a literature review section to clarify the positioning of this study with respect to the extant studies. It is essential to clarify why the demographic (independent) variables were expected to influence consumer buying decisions on healthy foods.

Response: Thanks. The writing format of this manuscript is exactly in accordance with the format provided by the Foods journal, and according to the format of the journal, there is no separate section called a literature review. Moreover, in many articles published in the “Foods” journal, the literature review section is mentioned in the introduction, and in this manuscript, the authors have preferred to follow this method and journal format.

The aim of this research was to investigate the factors affecting poultry meat consumption and then to investigate the factors affecting the consumption of ABF poultry meat. Since the factors affecting the consumption of poultry meat may be different from the factors affecting the consumption of ABF poultry meat, in this study two different models were used to investigate these two issues. In economic and marketing studies, various variables affect the consumption and purchase of different products, including healthy foods. Some of these variables, such as price or income, have been studied in various studies and are known variables. But some variables may have different effects from one region to another, and therefore a complete understanding of the geographical, economic and demographic characteristics of each region can lead to a better understanding of the factors affecting the consumption of these products. Previous studies have investigated factors affecting the consumption of organic or healthy products in the agricultural sector. For example, previous studies identified socio-economic such as education and age [27]; gender and occupation [28]; price and income [29]; product quality [30]; and consumer awareness and environmental considerations [31]. However, there is no study with the methodology employed in this research related to the consumption of ABF poultry meat.

Since the production and consumption of ABF poultry meat are expanding in different countries of the world, this research has first investigated the factors affecting the consumption of poultry meat and then investigated the factors affecting the consumption of ABF poultry meat.

Again, we thank the reviewers for their thoughtful comments. The comments and suggestions have helped improve the manuscript.

This manuscript is a resubmission of an earlier submission. The following is a list of the peer review reports and author responses from that submission.

Round 1

Reviewer 1 Report

The introduction is too wide. It could be shortened without losing any significance. I.E. lines 65-92 are just common knowledge and could be said in two sentences. Still, In the whole introduction, there is little argumentation regarding animal welfare and its influence on ATB usage on farms. 

lines 142-154 should be in Materials and methods.

In Materials and methods, the survey details should be presented separately - questions asked, why those questions, how many surveys - a type of surveying (online, line, face to face,..), and ideally - a full questionary attached as supplementary material. 

After that authors are welcome to discuss models used for data management, but I believe we can agree that methods of data collection should be presented ahead of that. 

Depending on the survey method we can discuss of 360 samples are enough for significant conclusions. 

Results are presented in a logical and satisfactory form, but there is a lot of discussion in them. So either collate the Discussion text with the results completely or divide it and build a stronger Discussion section. 

The conclusions are too broad. Authors should be making conclusions based only on their research not compiling general opinions. 

Is there ethical approval for collecting the data? 

Authors should consider inserting a paragraph regarding the weaknesses of their research and how they have overcome them. 

Author Response

Responses to the Comments from Reviewer #1

Comments and Suggestions for Authors

Comment 1: The introduction is too wide. It could be shortened without losing any significance. I.E. lines 65-92 are just common knowledge and could be said in two sentences. Still, in the whole introduction, there is little argumentation regarding animal welfare and its influence on ATB usage on farms. 

Response: We have addressed that now in the revised manuscript. Thanks!

Comment 2: In materials and methods, the survey details should be presented separately - questions asked, why those questions, how many surveys - a type of surveying (online, line, face to face,..), and ideally - a full questionaries’ attached as supplementary material. 

Response: We have clarified and added more explanations to address these issues in the revised manuscript.

Comment 3: After that authors are welcome to discuss models used for data management, but I believe we can agree that methods of data collection should be presented ahead of that. Depending on the survey method we can discuss of 360 samples are enough for significant conclusions. 

Response: We agree with the reviewer. Thanks!

Comment 4: Results are presented in a logical and satisfactory form, but there is a lot of discussion in them. So either collate the discussion text with the results completely or divide it and build a stronger discussion section. 

Response: Done. Thanks!

Comment 5: The conclusions are too broad. Authors should be making conclusions based only on their research not compiling general opinions. 

Response: Done. Thanks!

Comment 6: Is there ethical approval for collecting the data? 

Response: There are university procedures we must follow. In the present research, we followed all the university rules and regulations and did not allow our personal opinions to interfere with the data collection process.

Comment 7: Authors should consider inserting a paragraph regarding the weaknesses of their research and how they have overcome them. 

Response: We have addressed this issue in the discussion section in the revised manuscript.

Reviewer 2 Report

1. From the title you would realize that it has been a very interesting and important study because Antibiotic-free poultry meat consumption is a catchy topic and all over the world its now a serious issue but in this study the authors investigated the factors affecting poultry meat consumption and ABF poultry meat preference by consumers through a questionnaire method not through lab experiments.

2.  The topic is original and relevant to the field, but it is totally based on mathematical equations, there is no proper materials and methods in it, and it is totally out of scope of FOODS, I would recommend the authors to support their mathematical work with some experiments in the lab and then submit it to FOODS. The article has several flaws in it and a lot of grammatical and topographical mistakes. This article needs further additional experiments, research not conducted correctly.

3.  Experiments in the lab are required, the authors should read and follow the following published articles,

https://doi.org/10.3390%2Ffoods11030249

https://doi.org/10.3390/biology9110411

https://doi.org/10.3389/fnut.2021.692839. eCollection 2021.

https://doi.org/10.3390/ antibiotics11091222

https://doi.org/10.3382/japr/pfv006

https://doi.org/10.3390/ ani12182310

4.Only two models have been included in this study, Improved methodology is required.

Author Response

Comment 1: From the title you would realize that it has been a very interesting and important study because Antibiotic-free poultry meat consumption is a catchy topic and all over the world it’s now a serious issue but in this study the authors investigated the factors affecting poultry meat consumption and ABF poultry meat preference by consumers through a questionnaire method not through lab experiments.

Response: We agree with the reviewer. In economic and marketing studies of agricultural products, our laboratory is the society and consumers of the products, and our samples are part of the population with the characteristics of that society.

 Comment 2: The topic is original and relevant to the field, but it is totally based on mathematical equations, there is no proper materials and methods in it, and it is totally out of scope of FOODS, I would recommend the authors to support their mathematical work with some experiments in the lab and then submit it to FOODS. The article has several flaws in it and a lot of grammatical and topographical mistakes. This article needs further additional experiments, research not conducted correctly.

Response: In the economics methodology, econometrics and mathematical modeling are part of materials and methods, and the main tools of any scientific inquiry. Specifically, in economic and marketing studies, the two models employed in this study (Ordered logit and Tobit) are among the best and most common models for investigation of consumer behavior. In addition, studies regarding consumption of different foods are included in the subject areas of the Foods journal. Particularly, the topic studied in this research is directly related to the Special Issue: “Advances in Sustainable Agri-Food Systems: Insights into Production, Processing, and Consumption Perspectives

Comment 3: Experiments in the lab are required, the authors should read and follow the following published articles,

https://doi.org/10.3390%2Ffoods11030249

https://doi.org/10.3390/biology9110411

https://doi.org/10.3389/fnut.2021.692839. eCollection 2021.

https://doi.org/10.3390/ antibiotics11091222

https://doi.org/10.3382/japr/pfv006

https://doi.org/10.3390/ ani12182310

Response: We have reviewed some of that literature that are related to this research in the revised manuscript. Also please see our response to comment #2.

Comment 4: Only two models have been included in this study, improved methodology is required.

Response: As explained above, the two models used in this research are among the best commonly used models in economic and marketing studies. Furthermore, we think the methodology employed in this research is quite strong and scientific, used in many related books and articles. Interestingly, one of the authors referenced in our study has a book on this subject matter (reference 36 in the revised manuscript).

Reviewer 3 Report

The study's title looks attractive "Antibiotic-free poultry meat consumption and its determinants."

Data for this study were collected from questionnaires completed by 360 households, 238 heads in Mashhad, Iran, in 2021. More than the data from a single city is needed for such a study. At least 4 cities of different regions should be studied for a survey-based study to draw any meaningful conclusion regarding the demand and consumption of antibiotic-free poultry meat.

The biggest future challenge for a nutritionist is to fulfill the protein requirements of the world's increasing population. Chicken is the cheapest available source of high-quality proteins. It is also established that poultry (broilers) is more prone to diseases and stress than other animals. It is challenging to produce large-scale ABF chicken to fulfill the global demand. So, such a study can create some distress among the consumers of developing and developing countries, who are dependent only on cheaper Chicken to feed their population.

Why not add other ways to reduce antibiotic resistance rather than moving towards 100% organic chicken?

The best possibility is implementing a minimum 1-2 weeks antibiotic withdrawal period before slaughtering poultry (broiler).

I recommend modifying this manuscript so that it can not result in the downfall of the world's broiler industry. Because in its current form, it can cause panic among regular broiler users.  

Author Response

Comment 1: The study's title looks attractive "Antibiotic-free poultry meat consumption and its determinants." Data for this study were collected from questionnaires completed by 360 households, 238 heads in Mashhad, Iran, in 2021. More than the data from a single city is needed for such a study. At least 4 cities of different regions should be studied for a survey-based study to draw any meaningful conclusion regarding the demand and consumption of antibiotic-free poultry meat.

Response: Mashhad is the second largest city in Iran, has more than 3 million people population with a lot of recent diverse immigrants from different cities and villages all over Iran. This city has 13 municipal districts with different economic and social characteristics. Hence, taking a sample from this city with a large income and economic diversity can be considered as an alternative to sampling from several smaller cities.

Comment 2: The biggest future challenge for a nutritionist is to fulfill the protein requirements of the world's increasing population. Chicken is the cheapest available source of high-quality proteins. It is also established that poultry (broilers) is more prone to diseases and stress than other animals. It is challenging to produce large-scale ABF chicken to fulfill the global demand. So, such a study can create some distress among the consumers of developing and developing countries, who are dependent only on cheaper Chicken to feed their population.

Response: The issue in this research is examining the factors affecting consumer demand for ABF poultry meat. Consumers have different income levels and tastes and preferences, and for example, with an increase in income, demand for healthy products increases. Hence, with an increase in per capita income in various developed and developing countries, it is expected that the consumption of healthy products such as ABF poultry meat will increase. This research provides an appropriate guidance for producers given preferences of consumers. We agree with the reviewer that poultry (broilers) is more prone to diseases and stress than other animals, and a large population of the world depending on chicken meat that is cheaper to some extent. But this issue would be a concern worthy of attention and needs strategic response by agribusiness leaders and mangers.

Comment 3: Why not add other ways to reduce antibiotic resistance rather than moving towards 100% organic chicken? The best possibility is implementing a minimum 1-2 weeks antibiotic withdrawal period before slaughtering poultry (broiler).

Response: As mentioned on page 3 of the revised manuscript, organic poultry is ABF, but not all ABF is organic. The use of antibiotics is prohibited in both ABF poultry and organic poultry, but no other chemical additives are allowed in organic poultry throughout their production process. Organic broiler meat has variable costs that are 70% higher and fixed costs that are 86% higher than conventional production. This higher production cost for organic chicken is due to feed, labor, certification, and outdoor maintenance costs. ‎In this research, only the issue of antibiotic-free poultry was examined.

Comment 3: I recommend modifying this manuscript so that it cannot result in the downfall of the world's broiler industry. Because in its current form, it can cause panic among regular broiler users.  

Response: The production and consumption of ABF poultry meat is expanding in different countries of the world and such concerns must be addressed. In this research, we focused on investigating the factors affecting the consumption of poultry meat in general, and then the factors affecting the consumption of ABF poultry meat. So, this research investigated the preferences of consumers regarding a healthy product (i.e., ABF poultry meat). Future studies need to address the strategic response of producers and marketers to poultry meat safety concerns.

Round 2

Reviewer 2 Report

The authors have not answered my comments in a satisfactory way. My recommendation is the same that this article is not suitable for publication in FOODS. My recommendation is reject.

Reviewer 3 Report

The authors comprehensively addressed the questions and made necessary changes. I'm fully satisfied.